# Biomarker Testing in Older Patients Treated for an Advanced or Metastatic Non-Squamous Non-Small-Cell Lung Cancer: The French ESME Real-Life Multicenter Cohort Experience

**DOI:** 10.3390/cancers14010092

**Published:** 2021-12-24

**Authors:** Tina Lamy, Bastien Cabarrou, David Planchard, Xavier Quantin, Sophie Schneider, Michael Bringuier, Benjamin Besse, Nicolas Girard, Christos Chouaid, Thomas Filleron, Gaëtane Simon, Capucine Baldini

**Affiliations:** 1Département D’Oncologie Médicale, Gustave Roussy, 94805 Villejuif, France; david.planchard@gustaveroussy.fr (D.P.); benjamin.besse@gustaveroussy.fr (B.B.); 2Unité de Biostatistiques, Institut Claudius Regaud—IUCT-O, 31059 Toulouse, France; Cabarrou.Bastien@iuct-oncopole.fr (B.C.); Filleron.Thomas@iuct-oncopole.fr (T.F.); 3Institut de Recherche en Cancérologie de Montpellier (IRCM), Inserm U1194, Université de Montpellier, Institut du Cancer Montpellier (ICM), 34090 Montpellier, France; xavier.quantin@icm.unicancer.fr; 4Service de Pneumologie, Centre Hospitalier de la Côte Basque, 64100 Bayonne, France; sschneider@ch-cotebasque.fr; 5Département D’Oncologie Médicale, Institut Curie, 75005 Paris, France; michael.bringuier@curie.fr (M.B.); nicolas.girard2@curie.fr (N.G.); 6Service de Pneumologie, Centre Hospitalier Intercommunal de Créteil, 94000 Creteil, France; christos.Chouaid@chicreteil.fr; 7Direction des Datas, Unicancer, 75654 Paris, France; g-simon@unicancer.fr; 8Département D’Innovation Thérapeutique et des Essais Précoces (DITEP), Gustave Roussy, 95805 Villejuif, France

**Keywords:** lung cancer, biomarker testing, older patients

## Abstract

**Simple Summary:**

Genomic and immunologic tumor biomarker testing has dramatically changed the prognosis of patients treated for advanced/metastatic non-squamous non-small-cell lung cancer (aNSCLC). In older patients, targeted therapy and immunotherapy appear attractive considering better tolerance and increased survival. However, it remains unclear whether they have access to biomarker testing techniques in the same proportion as younger patients. The aim of our retrospective study was to compare the proportion of biomarker testing performed in non-squamous aNSCLC at diagnosis between patients aged ≥70 years old and their younger counterparts. There was no significant difference between the two age groups in terms of frequency of biomarker testing. Among old patients tested, 22% of them presented an *EGFR* mutation. Biomarker testing is a crucial diagnostic tool for older patients with aNSCLC in whom the newer anti-EGFR agents have shown clear benefits.

**Abstract:**

Background: Genomic and immunologic tumor biomarker testing has dramatically changed the prognosis of patients, particularly those treated for advanced/metastatic non-squamous non-small-cell lung cancer (aNSCLC) when access to targeted agents is available. It remains unclear whether older patients have access to therapy-predictive biomarker testing techniques in the same proportion as younger patients. This study aims to compare the proportion of biomarker testing performed in non-squamous aNSCLC at diagnosis between patients aged ≥70 years old and their younger counterparts. Methods: We conducted a retrospective analysis using the Epidemio-Strategy and Medical Economics (ESME) Advanced or Metastatic Lung Cancer Data Platform, a French multicenter real-life database. All patients with non-squamous aNSCLC diagnosed between 2015 and 2018 were selected. Biomarker testing corresponded to at least one molecular alteration and/or PD-L1 testing performed within 1 month before or 3 months after the aNSCLC diagnosis. Results: In total, 2848 patients aged ≥70 years and 6900 patients aged <70 years were included. Most patients were male. The proportion of current smokers at diagnosis was higher in the <70 years group (42% vs. 17%, *p* < 0.0001). There was no significant difference in the proportion of biomarker testing performed between the two groups (63% vs. 65%, *p* = 0.15). *EGFR* mutations were significantly more common in the older group (22% vs. 12%, *p* < 0.0001) and *KRAS* mutations significantly more frequent in the younger group (39% vs. 31% *p* < 0.0001). The distribution of other driver mutations (*ALK, ROS1, BRAF* V600E, *HER2,* and *MET*) was similar across age. In the multivariable analysis, factors independently associated with biomarker testing were gender, smoking status, history of COPD, stage at primary diagnosis, and histological type. Conclusions: Age is not a barrier to biomarker testing in patients with aNSCLC.

## 1. Introduction

Lung cancer is the most common cancer worldwide and the first leading cause of cancer-related death with 2.2 million new cases and 1.8 million deaths in 2020 [1]. The median age at diagnosis is 71 y and 72 y at death, for both sexes. Due to the aging population and longer life expectancy, lung cancer incidence is expected to double in the next thirty years [1]. 

Non-small-cell lung cancer (NSCLC) accounts for 80–90% of lung cancers. During the past two decades, lung cancer rates have decreased in men but are continuing to increase in women. These evolutions are explained, to some extent, by changes in lifestyles such as the tobacco consumption increase among women [2,3]. However, the histological features of NSCLC have drastically changed with a decreased incidence of squamous cell carcinomas concomitant to an increased incidence of adenocarcinomas. 

Among the population aged 70 years old or above, lung cancer is ranked among the four most frequent cancers diagnosed in the world. Higher mortality rates are observed in this population and are associated with a later stage at diagnosis with 60% of advanced or metastatic disease, and age-related comorbidities, limiting the use of systemic treatments such as platinum-based chemotherapy [4,5,6,7]. Few data exist concerning the distribution of histological subtypes among older patients in developed countries. Differences could be expected considering specific characteristics inherent to the older population such as a predominantly female population with lower exposure to tobacco, longer exposition to other carcinogens, and senescence. 

The advances in understanding genomic alterations in NSCLC and immune checkpoint inhibitory pathways led to the development of new drugs and the identification of predictive biomarkers of response to targeted therapy and immunotherapy. Molecular and PD-L1 expression testing now allows an individually tailored approach and improved survival. Molecular alterations include epidermal growth factor receptor (*EGFR*) gene mutations, anaplastic lymphoma kinase (*ALK*) gene rearrangements, ROS proto-oncogene 1 receptor tyrosine kinase (*ROS1*) gene mutations or rearrangements, B-Raf proto-oncogene serine/threonine kinase (*BRAF*) V600E gene mutations, KRAS proto-oncogene GTPase (*KRAS*) gene mutations, Human epidermal growth factor receptor 2 (*HER*2) gene mutations and amplifications, MET proto-oncogene receptor tyrosine kinase (*MET*) gene mutations or amplification, Ret proto-oncogene (*RET*) gene rearrangements, and neurotrophic receptor tyrosine kinase (*NTRK*) gene rearrangements. Targeted therapy and immunotherapy have drastically changed the prognosis of patients treated for an advanced/metastatic NSCLC to the point that, nowadays, they have become the standard of care in the front-line setting in patients with targetable genetic alterations and no contraindications to PD-1/PD-L1 inhibitors [8,9,10,11,12,13,14]. 

In older patients, targeted therapy and immunotherapy appear to be attractive options considering better tolerance and increased survival [15,16,17,18,19]. Indeed, studies of aNSCLC treatment in this particular population, although few in number and consisting mainly of retrospective analyses of subgroups of larger studies, have shown similar results to those of the general population in terms of progression-free survival, the objective response rate, and quality of life compared to chemotherapy. Yet, managing cancer in this population remains a challenge, from tissue biopsy for diagnosis to managing treatment-related toxicities to determining treatment strategy. All of this requires consideration of the patient’s general health, life expectancy, cognition, and preferences. All these aspects may discourage the clinician from going further in the diagnostic strategy, and especially in the search for molecular biomarkers. This study aims to compare the proportion of tumor biomarker testing performed in advanced or metastatic non-squamous NSCLC at diagnosis between patients aged ≥70 years old, and their younger counterparts. 

## 2. Materials and Methods

### 2.1. Data Sources–Data Extraction and Risk of Bias Assessment—ESME Data Platform and Study Population 

The ESME-Advanced or Metastatic Lung Cancer (ESME-AMLC) data platform of Unicancer is an ongoing unique national real-life cohort (clinicaltrials.gov NCT03848052), derived from electronic health records of consecutive patients treated for advanced or metastatic lung cancer since 2015 at one of the health facilities (18 private non-profit Comprehensive Cancer Centers and 18 University or General Hospitals). These facilities are selected to be representative of the French healthcare system for the treatment of advanced or metastatic lung cancer. Data are updated annually and run through a data management process aimed at ensuring the quality of the data analyzed. This database compiles data from patients’ electronic medical records, inpatient medical records, and medication records. In compliance with French regulations, the ESME-AMLC database was authorized by the French data protection authority. Unicancer manages the data platform in accordance with the current best practice guidelines. No formal dedicated informed consent was required, but all patients had approved the analysis of their electronically recorded data.

Patients were selected in the ESME database according to specific criteria: Aged 18 and over, with advanced or metastatic lung cancer treated in a medical center between 1 January 2015 and 31 December 2018. For the present study, data were collected until the cut-off date on 3 September 2019.

### 2.2. Data Collection

The following data were collected: Age at NSCLC diagnosis; sex; Eastern Cooperative Oncology Group Performance Status (0–2 vs. >2); smoking history (never, former, or current smoker); medical history including other cancer history, familial history of cancer, stage at primary diagnosis, metastatic status at primary diagnosis, histological subtype (adenocarcinoma vs. other), and metastatic first-line treatment.

### 2.3. Molecular Analysis

Since PD-L1 appeared later in the molecular exploration panel, we distinguished molecular alteration and PD-L1 expression testing. Biomarker testing was defined as at least one molecular alteration and/or PD-L1 testing performed within 1 month before or 3 months after the aNSCLC diagnosis. « Molecular testing only » meant testing for at least one molecular alteration excluding the search for PD-L1 status. Among the molecular alteration testing listed in the database, we analyzed *EGFR* mutation, *ALK* translocation, *KRAS* mutation, *BRAF*V600E mutation, *ROS*1 translocation and rearrangement, *HER*2 mutation, and *MET* mutation and amplification. Biomarker testing results were specified as « not done », « positive », « negative », or « not contributive ». Data such as the biopsy location and sample type, location and type of mutation, were not collected.

### 2.4. Outcomes

The primary objective of this study was to compare the proportion of biomarker testing performed in non-squamous aNSCLC at diagnosis between patients aged ≥70 years old and their younger counterparts.

The secondary objectives were to describe the frequency of the molecular alteration according to age group, and to identify factors associated with biomarker testing.

### 2.5. Statistical Analysis

Data were summarized by median and range for continuous variables and by frequency and percentage for categorical variables. The number of missing data was presented for each variable, but not considered for the percentage calculations. Comparisons between groups were performed using the Mann–Whitney test for continuous variables and the Chi-squared or Fisher’s exact test for categorical variables. A multivariable logistic regression model was performed to study the factors associated with biomarker testing. Odds ratios (OR) were estimated with their 95% confidence intervals (95% CI). All statistical tests were two-sided. Statistical analyses were conducted using STATA 16 software (StataCorp. 2019. Stata Statistical Software: Release 16. College Station, TX, USA: StataCorp LLC). 

### 2.6. Patient and Public Involvement

No patients were involved in setting the research question or the outcome measures, nor were they involved in developing plans for design and implementation of the study. No patients were asked to advise on interpretation or writing up of results. There are no plans to disseminate the results of the research to study participants or the relevant patient community. 

## 3. Results

### 3.1. Patient and Tumor Characteristics 

A total of 21,169 patients were included in the ESME-Advanced and Metastatic Lung Cancer database. Among them, 13,219 patients presented non-squamous NSCLC, including 9748 with an advanced or metastatic disease diagnosed between 1 January 2015 and 31 December 2018. Among patients with advanced or metastatic non-squamous NSCLC, there were 6900 (71%) aged under 70 years old (yo) and 2848 (29%) aged of 70 yo and above. Altogether, 6285 patients (64%) had biomarker testing at diagnosis (Figure 1). 

Patient characteristics are presented in Table 1. The median age at aNSCLC diagnosis was 64 yo (range 21–97 y). Most of the patients were male (63%) and non-smokers or former smokers (65%). Fifteen percent of patients reported a family history of lung/pleural tumors. Of the 3581 patients with recorded ECOG PS, 89% had an ECOG PS of 0–2. The most frequent histological subtype was adenocarcinoma in 8074 patients (83%), and other subtypes of non-squamous NSCLC included neuroendocrine large cell (3.3%), undifferentiated (3.5%), and other (10.4%). The stage at primary diagnosis was IIIB or IV in 79% of patients. A first-line metastatic treatment was delivered in 81% of patients.

Comparing the older and younger groups (Table 1), the distribution of men and women was similar between the two age groups (62% of men and 38% of women patients in the <70 yo group and 65% and 35% in the ≥70 yo group, respectively), the proportion of current smokers at diagnosis was higher in the younger group (42% vs. 17%, *p* < 0.0001) and, as expected, older patients had more medical history than their younger counterpart including chronic obstructive pulmonary disease (COPD) (19% vs. 15%, *p* < 0.0001), kidney failure (5% vs. 1%, *p* < 0.0001), heart failure (4% vs. 2%, *p* < 0.0001), diabetes mellitus (18% vs. 10%, *p* < 0.0001), high blood pressure (51% vs. 27% *p* < 0.0001), and other cancer history (28% vs. 14%, *p* < 0.0001). ECOG PS was poorer (>2) in the oldest group (14% vs. 10%, *p* = 0.0002). Younger patients were more frequently metastatic at primary diagnosis (72% vs. 67%, *p* < 0.0001) and had more family history of lung or pleural tumor (17% vs. 11% *p*= 0.0001). There was no difference in terms of a histological subtype with 83% of adenocarcinoma in both groups.

### 3.2. Biomarker Testing

Tumor biomarker testing (at least one molecular alteration and/or PD-L1 testing) performed at non-squamous aNSCLC diagnosis was performed in 64% of patients over the entire period studied. This proportion increased over time in both groups. In 2015 and 2016, a « molecular testing only » was performed in the majority of patients (59% and 58%, respectively) regardless of age, while « PDL1 status research only » was hardly ever performed (0.1% and 0.6%, respectively). Then, while « molecular testing only » was performed in 32% in 2017 and 8% in 2018, it began to be associated with the search for PD-L1 status in 34% and 54% of cases, respectively (molecular testing + PDL-1) (Figure 2). 

Analyzing each molecular alteration in detail, the proportion in which they were executed was the same over the years (Figure 3).

Comparing patients screened and not screened (Table 1), those who had tumor biomarker testing were more likely to be female (40% vs. 33%, *p* < 0.0001), to have an ECOG PS ≤ 2 (90% vs. 86%, *p* = 0.0009), to be non-smoker (15% vs. 8%, *p* < 0.0001), and without a large medical history (53% vs. 44%, *p* < 0.0001). There was no statistically significant difference in the proportion of biomarker testing performed between the two age groups (63% vs. 65%, *p* = 0.1593). Supplementary analysis with an increased age cutoff for old patients (≥75 years old and ≥80 years old, respectively) showed no significant difference in younger patients in biomarker testing at aNSCLC diagnosis, neither between patients aged ≥75 years and their younger counterparts nor between patients aged ≥80 years and younger (64.0% vs. 64.6% *p* = 0.6909 and 64.0% vs. 64.5% *p* = 0.7486, respectively) (Appendix A).

### 3.3. Molecular Alteration Characteristics According to Age Group

The frequency of each molecular alteration performed at advanced or metastatic non-squamous NSCLC diagnosis is shown (Table 2). The most frequent alterations tested were EGFR, ALK, and KRAS. EGFR testing was conducted in 53% of patients and was positive in 15% of cases. EGFR mutation was significantly more common in the older group (22% vs. 12%, *p* < 0.0001). ALK testing was performed in 46% of patients and was positive in 4% of cases. There was no difference between the two groups of age. KRAS testing was performed in 46% of patients and was positive in 37% of cases. KRAS mutations were significantly more frequent in the younger group (39% vs. 31%, *p* < 0.0001). Regarding the other molecular alterations, their distribution was similar from one group to another. 

### 3.4. Factors Associated with Biomarker Testing

In the multivariable analysis (Table 3), factors independently associated with biomarker testing were female gender (OR = 1.26; 95%CI = [1.13–1.41], *p* < 0.0001), adenocarcinoma histological type (OR = 2.36; 95%CI = [2.08–2.67], *p* < 0.0001), and stage IIIB–IV at primary diagnosis (OR = 7.56; 95%CI = [6.69–8.53], *p* < 0.0001). Medical history such as smoking status and history of COPD were independently associated with no testing (OR = 0.56; 95%CI = [0.46–0.68], *p* < 0.0001 and OR = 0.68; 95%CI = [0.59–0.78], *p* < 0.0001, respectively). Age did not emerge as an independently associated factor with one practice or another.

## 4. Discussion

To the best of our knowledge, this is the first study to focus on the real-life practice of biomarker testing in the older population with advanced or metastatic non-squamous NSCLC.

Over the study period, older patients with a diagnosis of advanced or metastatic NSCLC benefited from a tumor genotyping and PD-L1 testing status as much as their younger counterparts. In our multivariable analysis, factors associated with testing were female gender, adenocarcinoma, non-smoker status, and stage IIIB–IV at primary diagnosis. These factors are in accordance with current guidelines [14]. Although we know that the diagnosis of lung cancer in very old patients is challenging, from the safe acquisition of tissue to the determination of the treatment strategy and management of treatment-related toxicities, in our study, once a histological diagnosis has been made, the diagnostic process was completed by biomarker testing in 65% of cases, regardless of age.

This lack of significant difference according to age can be explained by the relatively young age of the geriatric population in our study (median age 76 yo) and a good ECOG PS (90% ECOG PS 0-2 among the 3581 patients with known data). Older patients selected were probably fit enough to be offered a comprehensive diagnostic strategy. 

Regarding the evolution of practices, tumor biomarker testing has become a common practice during the period studied with a clear democratization of the search for PD-L1 status from 2017. This is in accordance with the latest guidelines published in 2018, which strongly recommended, during the diagnosis of advanced or metastatic NSCLC, in addition to PD-L1, the search for EGFR mutations and rearrangements involving genes such as ALK and ROS1 and also encouraged screening for other alterations such as BRAF, HER2, MET, RET, and NTRK [14]. Our study shows that older patients have not remained on the sidelines of these therapeutic advances since they were offered the realization of a molecular profile in the same way as the youngest.

However, despite this significant evolution, while more than 80% of patients had stage IIIB or IV adenocarcinoma, only 65% of them had undergone testing, and this was regardless of age. In our study, we analyzed demographic, clinical, and histological factors associated with the implementation of biomarker testing, but firstly, we considered only testing performed within 3 months after the aNSCLC diagnosis, and secondly, we did not evaluate extrinsic factors such as the lack of tumor material, the technique used to perform the molecular analysis, or even the type of institution that referred the patient and its specific access to molecular testing laboratories. Further studies are warranted to describe barriers to testing in older adults in everyday clinical practice. In a recent international survey to evaluate perceptions on the current practice and barriers to the implementation of molecular and PDL-1 testing, Smeltzer and al. listed the top five barriers identified: Cost, quality and standards, access, and turnaround time [20]. These factors are consistent with other reports [21,22].

In France, Barlesi and al. demonstrated that routine nationwide molecular profiling of patients with advanced NSCLC was feasible, with acceptable turnaround times but with a limited number of genetic alterations (additional molecular alterations often tested only by research programs) [22]. However, they questioned the cost of such tools, considering that reimbursement policies have been changing, limiting access to molecular profiling. The cost effectiveness of genomic profiling has been studied in adenocarcinoma NSCLC but remains to be evaluated in France [22,23]. It could be one of the major challenges in the development of a comprehensive molecular sequencing tool accessible to everyone outside of clinical trials. 

In their study, Barlesi and al. found at least one potentially actionable molecular alteration in almost 50% of the analyses, leading to a specific treatment decision for 51% of patients. In our study, focusing on non-squamous aNSCLC, EGFR, ALK, and KRAS were the most frequent alterations tested with a frequency somewhat similar to what was previously reported [21,22,24,25]. We found significantly more EGFR mutations and fewer KRAS mutations in the group of patients aged 70 yo and above compared to the younger group. These results are consistent with those of Forest et al. who found a similar distribution across age in a French retrospective cohort, with a predominance of exon 19 deletion and L858R, known to be associated with better overall survival [25]. In our study, the type of EGFR mutation (exon 19 deletion, L858R, L861Q, and G719X) was not collected and should be further studied.

EGFR mutations are known to be associated with female gender, adenocarcinoma histology, and non-smoking status. The increased incidence of EGFR mutation in the older population in our cohort is consistent with the characteristics of the population i.e., mainly non-smoker. These encouraging results highlight the feasibility and potential therapeutic impact of molecular testing in older patients. This attitude is supported by studies showing favorable outcomes of tyrosine kinase inhibitors in first- or second-line treatment in older or frail patients with EGFR mutation-positive non-small cell lung cancer [15,26,27,28,29].

Genomic profiling and PD-L1 testing are crucial tools to inform prognosis, support treatment decisions, and provide access to innovation. The identification of “druggable” oncogenes and actionable biomarkers and the application of a dedicated therapeutic strategy is now associated with prolonged overall survival among patients harboring the corresponding alteration [21,29]. Liquid biopsy, being non-invasive and accessible, could be a determinant tool to increase access to molecular profiling especially in the older population who have been proven to take advantage of new anti-EGFR agents [30,31]. 

## 5. Conclusions

This study showed that the realization of biomarker testing including tumor genotyping and PD-L1 status research during the diagnosis of advanced or metastatic non-squamous NSCLC has become widespread over time and is now a routine practice. Older patients who had access to these diagnostic techniques presented an EGFR mutation in more than 20% of cases. The elderly must be able to benefit from a comprehensive geriatric evaluation at diagnosis in order to optimize their access to such diagnostic tools as much as possible, which today lead to personalized medicine. 

## Figures and Tables

**Figure 1 cancers-14-00092-f001:**
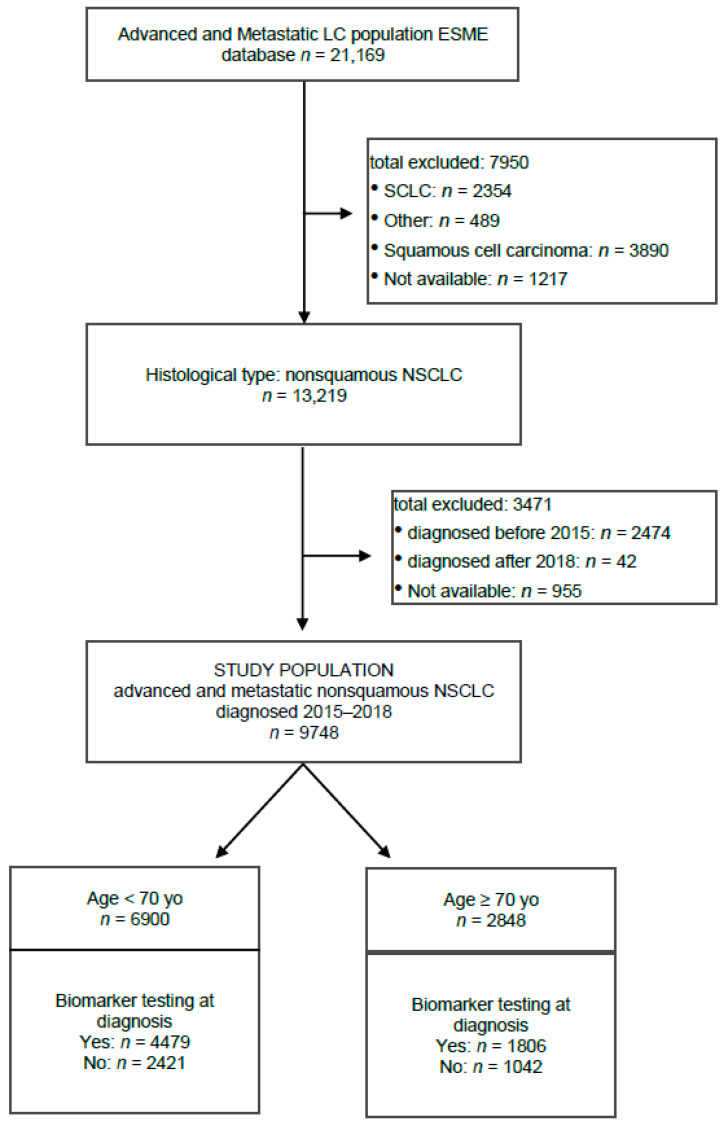
Study flow chart. LC: Lung cancer; SCLC: Small-Cell Lung Cancer; NSCLC: Non-Small-Cell Lung Cancer; yo: Years old.

**Figure 2 cancers-14-00092-f002:**
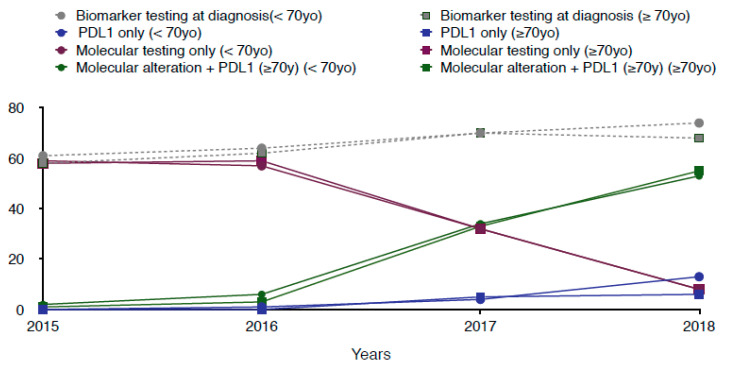
Molecular profiling over time according to age (<70 yo and ≥70 yo). Biomarker testing was defined as at least one molecular alteration and/or PD-L1 testing performed within 1 month before or 3 months after the non-squamous aNSCLC diagnosis. « Molecular testing only » meant testing for at least one molecular alteration excluding the search for PD-L1 status. Among molecular alteration testing listed in the database, we analyzed EGFR mutation, ALK translocation, KRAS mutation, BRAFV600E mutation, ROS1 translocation and/or rearrangement, HER2 mutation, and MET mutation and/or amplification.

**Figure 3 cancers-14-00092-f003:**
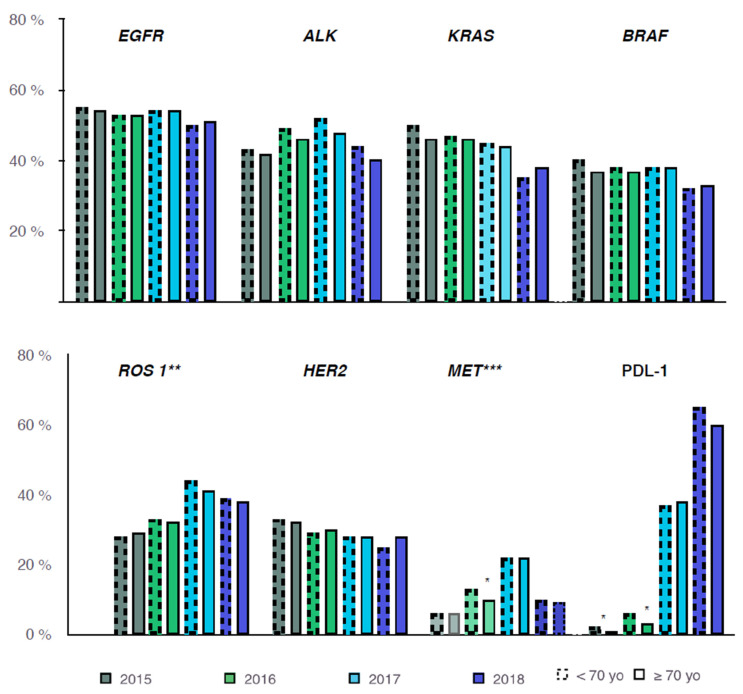
Biomarker testing rate according to age (<70 yo and ≥70 yo) over time. EGFR, epidermal growth factor receptor; ALK, anaplastic lymphoma receptor tyrosine; HER2, Human Epidermal growth factor Receptor 2; PDL-1, programmed cell death ligand 1. Positive result for PDL-1 meant a tumor proportion score >1%. «Molecular testing only», meaning testing for at least one molecular alteration excluding the search for PD-L1 status, were stable all over years. * *p* < 0.01, ** ROS1 mutation and/or rearrangement, *** MET mutation and/or amplification.

**Table 1 cancers-14-00092-t001:** Patient characteristics.

	All	<70y	≥70y	*p*	Tested	Non Tested	*p*
	*n* = 9748	*n* = 6900	*n* = 2848		*n* = 6285	*N* = 3463	
Median age at diagnosis, years (range) (*n* = 9748)	64 (21−97)	60 (21–69)	76 (70–97)	*0.0001*	64 (21–97) <70y: *n* = 4479 ≥70y: *n* = 1806	64 (29–94) <70y: *n* = 2421 ≥70y: *n* = 1806	*0.0195*
Sex (*n* = 9748)				*0.0014*			*0.0001*
Male	6111 (63%)	4256 (62%)	1855 (65%)		3780 (60%)	2331 (67%)	
Female	3637 (37%)	2644 (38%)	993 (35%)		2505 (40%)	1132 (33%)	
ECOG PS (*n* = 3581)				*0.0002*			*0.0009*
0–2	3183 (89%)	2385 (90%)	798 (86%)		2272 (90%)	911 (86%)	
>2	398 (11%)	264 (10%)	134 (14%)		252 (10%)	146 (14%)	
Missing	6167	4251	1916		3761	2406	
Smoker status (*n* = 9233)				*0.0001*			*0.0001*
No	1172 (13%)	577(9%)	595 (23%)		930 (15%)	242 (8%)	
Former	4813 (52%)	3227 (49%)	1586 (60%)		3039 (50%)	1774 (55%)	
Current	3248 (35%)	2790 (42%)	458 (17%)		2061 (34%)	1187 (37%)	
Missing	515	306	209		255	260	
Medical history (*n* = 9251)							
COPD	1504 (16%)	988 (15%)	516 (19%)	*0.0001*	741 (12%)	763 (24%)	*0.0001*
Heart failure	216 (2%)	113 (2%)	103 (4%)	*0.0001*	116 (2%)	100 (3%)	*0.0003*
Kidney failure	209 (2%)	82 (1%)	127 (5%)	*0.0001*	116 (2%)	93 (3%)	*0.0029*
DM	1167 (13%)	668 (10%)	499 (18%)	*0.0001*	710 (12%)	457 (14%)	*0.0008*
HBP	3156 (34%)	3156 (34%)	1760 (27%)	*0.0001*	2034 (34%)	1122 (35%)	*0.2795*
Missing	497	383	114		254	243	
Other cancer history (*n* = 9263)	1650 (18%)	886 (14%)	764 (28%)	*0.0001*	997 (17%)	653 (20%)	*0.0001*
Missing	485	368	117		280	205	
Family history of lung/pleural tumor (*n* = 4099)	619 (15%)	493 (17%)	126 (11%)	*0.0001*	409 (15%)	210 (15%)	*0.8963*
Missing	5649	3920	1729		3586	2063	
Histology (*n* = 9748)				*0.5918*			*0.0001*
Adenocarcin	8074 (83%)	5706 (83%)	2368 (83%)		5459 (87%)	2615 (76%)	
other	1674 (17%)	1194 (17%)	480 (17%)		826 (13%)	848 (25%)	
Stage at primary diagnosis (*n* = 9542)				*0.0001*			*0.0001*
I–IIIA	2022 (21%)	1295 (19%)	727 (26%)		596 (10%)	1426 (43%)	
IIIB–IV	7520 (79%)	5471 (81%)	2049 (74%)		5626 (90%)	1894 (57%)	
Missing	206	134	72		63	143	
Metastatic at primary diagnosis (*n* = 9542)	6699 (70%)	4840 (72%)	1859 (67%)	*0.0001*	5031 (81%)	1668 (50%)	*0.0001*
Missing	206	134	72		63	143	
First metastatic treatment line (*n* = 9748)	7909 (81%)	5908 (85%)	2001 (70%)	*0.0001*	5503 (88%)	2406 (70%)	*0.0001*

ECOG PS: Eastern Cooperative Oncology Group. COPD: Chronic Obstructive Pulmonary Disease. DM: Diabetes Mellitus. HBP: High Blood Pressure. Adenocarcin: Adenocarcinoma.

**Table 2 cancers-14-00092-t002:** Proportion of molecular alteration in each age group.

*n* tot = 9748 Patients	EGFR *n* = 5197 (53%)	ALK *n* = 4521 (46%)	KRAS *n* = 4448 (45%)	BRAF *n* = 3671 (38%)	ROS1 ** *n* = 3332 (34%)	HER2 *n* = 2921 (30%)	MET *** *n* = 1208 (12%)	PDL-1 *n* = 1664 (17%)
Contributive result	5082	4448	4324	3542	3273	2827	1183	1552
Positive result (%)	745/5082 (15%)	188/4448 (4%)	1588/4448 (37%)	175/3671 (5%)	76/3273 (2%)	47/2827 (2%)	160/1183 (14%)	929/1152 (60%)
<70 yo	425/3683 (12%)	135 (4%)	1206 *(39%) **	122 (5%)	50 (2%)	29 (1%)	122 (14%)	664 (60%)
≥70 yo	320/1514 *(22%) **	53 (4%)	382 (31%)	52 (5%)	26 (3%)	18 (2%)	38 (11%)	265 (59%)

EGFR, epidermal growth factor receptor; ALK, anaplastic lymphoma receptor tyrosine; HER2, Human Epidermal growth factor Receptor 2; PDL-1, programmed cell death ligand 1. * *p* < 0,01, ** ROS1 mutation and/or rearrangement, *** MET mutation and/or amplification.

**Table 3 cancers-14-00092-t003:** Factors associated with biomarker testing in multivariable analysis.

	OR	95%CI	*p*		OR	95%CI	*p*
Age at diag				Diabete Mellitus			
<70 yo	1.00			No	1.00		
≥70 yo	1.04	[0.92;1.17]	*0.56*	Yes	0.93	[0.80;1.09]	*0.38*
Gender				HBP			
Male	1.00			No	1.00		
Female	1.26	[1.13;1.41]	*<0.0001*	Yes	1.02	[0.91;1.14]	*0.73*
Smoker status				Other cancer history			
No	1.00						
Former	0.60	[0.50;0.72]	*<0.0001*	No	1.00		
Smoker	0.56	[0.46;0.68]	*<0.0001*	Yes	0.95	[0.83;1.09]	*0.47*
COPD				Histological type			
No	1.00			Other	1.00		
Yes	0.68	[0.59;0.78]	*<0.0001*	Adenocarcin	2.36	[2.08;2.67]	*<0.0001*
Kidney fail				Stage at primary diagnosis			
No	1.00			I–IIIA	1.00		
Yes	1.01	[0.72;1.42]	*0.95*	IIIB–IV	7.56	[6.69;8.53]	*<0.0001*
Heart fail							
No	1.00						
Yes	0.81	[0.58;1.13]	*0.2126*				

HBP: High Blood Pressure; COPD: Chronic Obstructive Pulmonary Disease; Adenocarcin: Adenocarcinoma.

## Data Availability

The datasets analyzed during the current study are available in the Epidemio-Strategy and Medical Economics (ESME) Advanced and Metastatic Lung Cancer (AMLC) Data Platform. The database of the ESME program or the database of the AMLC cohorts are currently not accessible. For any specific demand, please contact the corresponding author. Each demand will be examined on a case-by-case basis by the scientific committee.

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
