# Peer review of "Biomarker Testing in Older Patients Treated for an Advanced or Metastatic Non-Squamous Non-Small-Cell Lung Cancer: The French ESME Real-Life Multicenter Cohort Experience"

_cancers, 2021, doi:10.3390/cancers14010092_

Round 1

Reviewer 1 Report

Reply

The manuscript concerns the comparison of the proportion of biomarker testing performed in advanced/metastatic non-squamous non-small cell lung cancer at diagnosis between patients aged ≥ 70 years old and their younger counterparts.

The work is presented in approachable, but scientific manner. The manuscript is technically appropriate, and the data support the conclusions. There are few questions/suggestions to be proposed to enhance the manuscript:

  1. What is the type of sample (specimen)? Please include this information in Materials and Methods section since it is not obvious for the reader.
  2. Please provide a reference for choosing the statistical tools used in the study.
  3. Figure 2 is not clear and of low quality. The symbols from the right part of legend (crowns?) and left (diamonds?), they are not portrayed in the graph; instead of them it can be seen circles and triangles and even mixture of them in the shape of tear. Please correct the figure.
  4. Why the extrinsic factors were omitted? Please justify.
  5. Whether normality of data was tested? Any kind of post-hoc test was performed? Why Kruskal-Wallis test (non-parametric) was chosen? For Chi-squared or Fisher’s exact test percentage values were used as input? For this type of test should rather be used incidence values.
  6. The reported p values were adjusted using any method?

Based on article evaluation, I recommend Minor Revision of this manuscript.

Author Response

  1. What is the type of sample (specimen)? This is a real life cohort. As asked, this information has been updated in Materials and Methods section.
  2. Please provide a reference for choosing the statistical tools used in the study. Statistical analyses were carried out using STATA version 16 which is a software for statistics and data science. The citation of the Stata 16 software has been updated in the “Statistical analysis” section according to the recommendations of the Stata Technical Support (https://www.stata.com/support/faqs/resources/citing-software-documentation-faqs/)

  3. Figure 2 is not clear and of low quality. The symbols from the right part of legend (crowns?) and left (diamonds?), they are not portrayed in the graph; instead of them it can be seen circles and triangles and even mixture of them in the shape of tear. Please correct the figure. The figure has been modified to improve its visibility and readability, the triangles (which represented the data in ≥ 70 yo) have been replaced by squares, the circles (which represent the data in <70) are left as they are.
  4. Why the extrinsic factors were omitted? ESME-Advanced or Metastatic Lung Cancer (ESME-AMLC) data platform does not provide these data. These data are not collected.
  5. Whether normality of data was tested? Any kind of post-hoc test was performed? Why Kruskal-Wallis test (non-parametric) was chosen? For Chi-squared or Fisher’s exact test percentage values were used as input? For this type of test should rather be used incidence values. The normality of data was not tested and we have chosen to use a non-parametric test which does not rely on an assumption of distribution of the data. Comparison of continuous data between groups was performed using the Kruskal-Wallis test which is a multisample generalization of the two-sample Mann-Whitney rank-sum test. In our study, comparisons were made between two groups and the two tests are therefore equivalent in this setting. In order to avoid any ambiguity, this point was clarified in “Statistical analysis” section.
    No post-hoc tests were performed in this study.
    Comparison of categorical variables between groups was performed using the Chi-squared or Fisher’s exact test which are based on frequency observed/expected in each category.
  6. The reported p values were adjusted using any method? No adjustment of reported p-values was performed in this study.

Reviewer 2 Report

1. Table 1 and Figure 4 are unreadable and need to be redone. 2. The authors include non-squamous cell NSCLC, but isolate adenocarcinoma and others. What does the other subgroup include? It may be more expedient to limit ourselves to adenocarcinoma? 3. In general, I do not understand the purpose of the work. Why should older age limit the detection of biomarkers for lung cancer mutations? It is in this group of patients that there are often contraindications for surgical treatment and aggressive treatment with platinum-containing chemotherapeutic drugs, and the determination of biomarkers provides wider therapeutic possibilities. 4. How can the factors shown in Table 2 be related to the definition or definition of biomarkers?

Author Response

Dear Reviewer, thank you very much for your comments.

  1. Table 1 and Figure 4 have been redone for better readability.
  2. The authors include non-squamous cell NSCLC, but isolate adenocarcinoma and others. What does the other subgroup include? It may be more expedient to limit ourselves to adenocarcinoma? The most frequent histological subtype was adenocarcinoma in 83%, other subtypes of non squamous NSCLC included neuroendocrine large cell in 3,3%, undifferentiated in 3,5% and other in 10,4%. These data have been updated in the " Patient & tumor characteristics".
  3. Totally agree with this In general, I do not understand the purpose of the work. Why should older age limit the detection of biomarkers for lung cancer mutations? It is in this group of patients that there are often contraindications for surgical treatment and aggressive treatment with platinum-containing chemotherapeutic drugs, and the determination of biomarkers provides wider therapeutic possibilities. Totally agree with this comment. Older patients should be offered targeted therapy (for the reasons you mention), with pre-therapy molecular screening as a corollary. However, in real life, large-scale molecular screening can be a real challenge, so we wanted to see if, like younger patients, older patients have access to therapeutic innovation.
  4. How can the factors shown in Table 2 be related to the definition or definition of biomarkers? 

    In Table 2, factors such as diabetes mellitus, renal failure, COPD, and High Blood Pressure are factors, described in the literature, associated with polymorbidity and frailty in older patients.

    Then, factors such as smoking status, histological type, and gender, are factors associated with the presence of an activating oncogene mutation.

    These two types of confounding factors could therefore influence the choice of whether or not to perform a molecular testing.

Reviewer 3 Report

In the current manuscript the authors describe their findings regarding the age-related prevalence of testing in a national database of molecular testing in lung cancer from 2015-2018. They find that there no significant differences in testing frequency between those <70 and patients >70 years of age, but interesting discrepancies in the frequency of mutations found. The paper is well written but would benefit from meticulous proofreading to correct grammatical errors. There are also some specific points that could be addressed to strengthen its impact.

  1. In the summary the central finding of the paper, i.e. that testing rates do not differ between the age groups defined by the authors, should be mentioned. Instead they use the term 'benefited from' throughout the text, which is not correct and confusing and should be changed to something like 'underwent' or 'received', as benefit has not bene proven in this study.
  2. In line 68 they use 'predicted' for a date in the s past (2018); is this correct, i.e. are the data still to be analysed?
  3. In line 95 the describe HER2 and in line 97 they mention ERBB2. These are the same surely?
  4. As an explanation for a lack of difference in the testing frequency between the age groups, the authors in the discussion refer to the fact that the geriatric population defined in this study is relatively young. This is a valid point, and it would therefore be interesting to see whether a difference is seen if the cutoff is increased (i.e. to e.g. 75 or 80 years of age). 

Author Response

Dear Reviewer, thank you for your comments.

  1. the term "benefited from" has been changed for a more objective term;
  2. In line 68 they use 'predicted' for a date in the s past (2018); is this correct, i.e. are the data still to be analysed? Indeed, the article started being written in 2019... this is an editorial blunder, so the epidemiological data has been updated in the manuscript....
  3. In line 95 the describe HER2 and in line 97 they mention ERBB2. These are the same surely? Indeed,changes have been updated.
  4. As an explanation for a lack of difference in the testing frequency between the age groups, the authors in the discussion refer to the fact that the geriatric population defined in this study is relatively young. This is a valid point, and it would therefore be interesting to see whether a difference is seen if the cutoff is increased (i.e. to e.g. 75 or 80 years of age). 

    We performed subgroup analyses by age ( ≥75 years, and also  ≥80 years). These showed no significant difference from the younger population in terms of biomarker testing. These data have been updated in the section " biomarker testing", and will be provided in supplementary data.

Round 2

Reviewer 2 Report

The authors partially answered the questions posed. However, the tables are still included in the text of the manuscript as figures, therefore they are difficult to read and must be corrected. 

Author Response

Dear reviewer, thank you for your comment

To be consistent with the designations, "Figure 4: Proportion of molecular alteration in each age group" has been changed to "Table 2: Proportion of molecular alteration in each age group".

And therefore, to be consistent with the order of appearance of the tables in the article, "Table 2: Factors Associated with Biomarker Testing in Multivariable Analysis" became "Table 3: Factors associated with biomarker testing in multivariate analysis."